# The Impact of the COVID-19 Pandemic on Oncology Care and Clinical Trials

**DOI:** 10.3390/cancers13235924

**Published:** 2021-11-25

**Authors:** Jennyfa K. Ali, John C. Riches

**Affiliations:** Centre for Haemato-Oncology, Barts Cancer Institute, Queen Mary University of London, 3rd Floor John Vane Science Centre, Charterhouse Square, London EC1M 6BQ, UK; jennyfa.ali@rmh.nhs.uk

**Keywords:** SARS-Cov-2, COVID-19, oncology, cancer screening, clinical trials

## Abstract

**Simple Summary:**

The coronavirus pandemic has had a considerable impact on all parts of society. Unsurprisingly, healthcare has been particularly affected, including cancer care and trials of new drugs. This article will summarize the impact the pandemic has had on cancer healthcare taking into consideration how the pandemic affected potential cancer patients and stopped them seeking medical advice for new symptoms. The pandemic also affected the ability of people to access healthcare services and undergo the tests necessary to diagnose cancer. This article will also discuss the impact of the pandemic on existing treatments and the trials of new drugs. In light of the unprecedented speed of development of new treatments and vaccines for the virus itself, it will also review whether some of these adaptations could be used to accelerate the development of novel cancer therapies.

**Abstract:**

The coronavirus disease 2019 (COVID-19) pandemic has caused considerable global disruption to clinical practice. This article will review the impact that the pandemic has had on oncology clinical trials. It will assess the effect of the COVID-19 situation on the initial presentation and investigation of patients with suspected cancer. It will also review the impact of the pandemic on the subsequent management of cancer patients, and how clinical trial approval, recruitment, and conduct were affected during the pandemic. An intriguing aspect of the pandemic is that clinical trials investigating treatments for COVID-19 and vaccinations against the causative virus, SARS-CoV-2, have been approved and conducted at an unprecedented speed. In light of this, this review will also discuss the potential that this enhanced regulatory environment could have on the running of oncology clinical trials in the future.

## 1. Introduction

The emergence of the novel coronavirus severe acute respiratory syndrome coronavirus 2 (SARS-CoV-2) caused a global healthcare crisis that reshaped both standard cancer care practices and oncology research [1]. The reasons for this were multifactorial. Firstly, efforts to contain transmission led to the withdrawal of face-to-face consultations, the suspension of normal diagnostic services, and the rapid expansion of telemedicine, all with the aim of reducing viral spread from healthcare worker to the patient and vice versa [2]. Secondly, the impact of severe coronavirus disease 2019 (COVID-19) on healthcare systems led to a redeployment of healthcare workers to deal with the pandemic, which inevitably had an impact on cancer pathways. These factors also affected clinical trials, which were often initially suspended, before being restarted with substantial amendments to facilitate their conduct in the pandemic-era world. Furthermore, it was clear from initial reports that patients with significant co-morbidities, including cancer, were at increased risk of having severe disease and death [3,4]. Mortality rates for patients with cancer and symptomatic COVID-19 have been reported at 24–28% [5,6]. Consequently, cancer patients and healthcare professionals often faced difficult decisions regarding treatment, with the benefit of therapy having to be factored against the increased risk of acquiring SARS-CoV-2 from hospitalization/increased frequency of visits to hospital and developing more severe COVID-19 due to modulation of the immune system by therapy. This article will review the many impacts of COVID-19 on cancer care, from the initial point of entry into cancer care pathways through to the longer-term implications (Figure 1). In addition, it will focus on the impact on oncology clinical trials. Interestingly, implementing a more patient-centric approach in clinical trials somewhat inadvertently improved working efficiency, patient experience, and trial running costs. In addition, given the unprecedented speed with which COVID-19 clinical trials were approved and conducted, it will also review whether there are lessons that could be learnt to benefit the oncology community.

## 2. Impact on the Initial Presentation of Patients with Potential Cancer

The COVID-19 pandemic affected how patients were first suspected of having cancer in several different ways. Routine screening services were typically paused in the early stages of the pandemic over concerns over viral transmission and as staff were redirected toward COVID-19-directed care. Disruption to screening services decreases the potential for early cancer detection in many cases, particularly in routine screening of at-risk populations such as breast screening of women with high familial risk and colorectal cancer (CRC) screening for individuals with inflammatory bowel disease (IBD) or Familial Adenomatous Polyposis (FAP) [7,8,9,10]. In the United Kingdom (UK), it has been estimated that 3 million fewer people were screened between the months of March to September 2020. This resulted in a 42% reduction in the number of patients initiating treatment following diagnosis via screening in the year April 2020–March 2021 [11]. This raises the rather alarming question of what happened to all those missing cases: were they diagnosed in good time through alternative pathways or do they still remain undiagnosed? Therefore, a long-term consequence of the pandemic may well be that these missed diagnoses will be picked up later, indirectly adding to the morbidity and mortality from SARS-CoV-2 [12]. 

A further impact is reflected in the behavior of the common populace themselves. A common feature of many governmental responses to the pandemic was to enforce a widespread “lockdown”, essentially confining people to their homes. In the UK, a National Health Service (NHS) survey conducted in April 2020 to gain insight into public attitudes towards seeking medical advice for suspected cancer found that over 50% of the public had concerns and were hesitant about seeking help due to the pandemic. While shopping for essentials and seeking medical care were typically allowed, the overall impact of lockdown measures combined with the degree and nature of media coverage effectively discouraged people from seeking medical advice. Symptoms that may have normally provoked a visit to a primary care physician were not acted on, as people sought to adhere to the regulations in an attempt to avoid the virus or out of fear of burdening an already overwhelmed healthcare service [11]. In addition, there has also been concern that individuals were not reporting potential symptoms of cancer due to mis-attributing them to COVID-19. The prime example of this is lung cancer, where the development of respiratory symptoms such as a cough could be easily attributed to SARS-CoV-2 [13,14]. This is particularly worrying given the association of lung cancer mortality with more advanced stage presentation with metastatic disease [15]. Currently, there are little data to confirm an excess of lung cancer deaths due to this phenomenon, although this may change over the coming months and years. 

A further consideration is the impact of the pandemic on primary care services. Many general practitioners ceased face-to-face services altogether, which also included a significant reduction in basic frontline diagnostics such as blood tests and X-rays. Primary care services were also forced to adapt to prevent the spread of SARS-CoV-2, particularly to the elderly and vulnerable at greater risk of fatal COVID-19. Social contact was avoided wherever possible, resulting in a switch to the widespread usage of telemedicine platforms to replace in-person consultations [16]. It can be argued that deviation from the normal process is enough to deter people from seeking advice regarding their health concerns, particularly those with limited access/familiarity with online portals. It is also noteworthy that symptoms that would have been observed in a physical appointment may not have been noticed by the primary care physicians or may be subjectively downplayed by the patient during a telephone/video call appointment. 

All of these factors would have contributed to early cancer symptoms being missed, which was reflected in the reduction in referrals to secondary care. It has been widely noted that there was a significant decrease in the number of urgent referrals after the first lockdown was imposed in March 2020. The largest declines were observed specifically during the peaks of each wave of the pandemic. In England, NHS statistics recorded the largest fall in April 2020 when a 60% decrease in urgent referrals was recorded compared with April 2019 [17]. Overall, it was estimated that there have potentially been ~50,000 cases of undiagnosed cancer because of the disruption caused to normal primary care services and routine screening—equating to approximately 1 in 1000 of the UK population. However, the negative effects of COVID-19 on oncology care are not exclusively a consequence of the impact on primary care services and screening, but also have been significantly impacted by public healthcare seeking attitudes due to fear of exposure to the virus [11].

## 3. Impact on the Investigation of Patients with Suspected Cancer

In addition to the impacts on primary care, the COVID-19 pandemic has also had a massive impact on secondary care. Hospitals and health centers reduced their functioning capacity to maintain social distancing measures in accordance with government guidelines, which consequently delayed cancer diagnosis. Furthermore, any staff who displayed symptoms/tested positive for COVID-19 (or were exposed to someone who has tested positive) were also required to self-isolate as part of government guidelines, which further disrupted staffing levels and limited functioning capacity [18]. The UK NHS operational standard for the 2 week-wait urgent referral system states that 93% of patients with suspected cancer should be seen by a specialist within 14 days from primary care referral. This standard was not met in 2020/2021, when 88.7% of all urgent cancer referrals were seen within 14 days, representing a 2.8% decrease from the previous year (90.8%). Impact varied based on cancer type, therefore case burden should also be considered when considering the impact on diagnostic services. For example, suspected breast cancer referrals displayed the poorest performance with only 80.8% of urgent referrals meeting the 2-week timeframe. It is likely the larger case burden of breast cancer overwhelmed breast cancer services operating at reduced capacity due to the pandemic [19]. The reduced functioning capacity introduced a greater dependence on triaging patients in order of urgency, with resources being prioritized to cases with a high index of suspicion of cancer [20]. Subsequently, patients displaying experiencing milder symptoms who did not meet the criteria for urgent referral experienced the most significant delay in referral to secondary care [21]. The delays were not just confined to the initial assessment in secondary care, but were found throughout the pathway. For example, there was a significant reduction in surgical procedures such as biopsies, as surgeons, theatre staff, anesthesiologists, and ventilators were redeployed to assist with the pandemic [22]. A recent study conducted using data from Irish hospitals observed a 21.5% reduction in the number of biopsy procedures carried out between January–June of 2020 in comparison with the same time in 2019. When examined more closely, it is noted that the largest decline (−48%) was recorded between the period of April to June, which was soon after lockdown restrictions went into place in mid-March 2020 [23]. While diagnostic delays will clearly have an impact on the likelihood of successfully treating a particular tumor, a later diagnosis also causes significant mental and emotional stress [9]. This likely to have been particularly magnified due to the nature of the pandemic, with physical distancing contributing to the distress caused by removal from personal support systems such as family and friends [24]. 

## 4. Impact of the Management of Patients with Cancer

It was observed early in the pandemic that patients with cancer were at higher risk of developing severe and fatal COVID-19 [3,4]. This was due to cancers generally occurring more commonly in a higher risk population (e.g., the elderly), the immunosuppressive, and/or deconditioning effect of many malignancies, as well as the immune-modulatory effects of some treatments [25]. Somewhat surprisingly, large meta-analyses and registry studies have shown that systemic anticancer therapy per se does not increase the risk of dying from COVID-19. Instead, factors such as having cancer, older age, and the presence of other significant co-morbidities have been repeatedly demonstrated to be the major risk factors [25,26]. Several different strategies were adopted to try and deal with these problems. Risk−benefit analyses were typically conducted on a patient-by-patient basis, with treatments being postponed or withdrawn where the benefit offered by therapy was marginal [27,28,29]. Where possible, chemotherapy regimens were switched to the oral route to reduce the need for patients to attend hospital where they would run the risk of acquiring SARS-CoV-2 [30]. There were also recommendations made to supportive care, for example for the widespread use of granulocyte colony stimulating factor injections to reduce the risk of hospitalization from febrile neutropenia [31]. Consequently, more stringent measures were taken to shelter this population from contracting the virus and many scheduled treatments (i.e., chemotherapy, radiotherapy, and surgery) during the beginning of the pandemic were postponed while adaptive approaches of delivering therapy were established [32,33]. Cancer services rapidly developed strategies to overcome this setback and continue cancer care while minimizing the risk of COVID-19 transmission. COVID free hubs were set-up, distinct from hospitals, where cancer patients could receive their treatment with a reduced risk of virus transmission [34]. This included the use of “chemo buses” that could be parked on site of hospitals or moved to a more convenient location for patients. This effectively minimized the exposure to SARS-CoV-2 transmission in clinical settings while receiving therapy.

In the UK, NHS operational standards for new primary cancers state that treatment should commence within 31 days for 98% of patients receiving chemotherapy or other anti-cancer drugs, and for 94% of patients for radiotherapy and surgery. The overall performance for anti-cancer drugs and radiotherapy in 2020/2021 was above operational standards, which shows that services were able to adapt and continue with these treatments. However, despite efforts to maintain service levels, surgery was more significantly impacted during the pandemic and failed to meet operational standards, with only 88% of patients receiving surgery within 31 days [19]. Despite measures enforced to prevent SARS-CoV-2 infection, patients themselves also chose to delay their treatment in fear of contracting the virus by leaving self-isolation to receive treatment. One retrospective study of 165 lung cancer patients found that 9.1% of patients experienced a delay in receiving their treatment, although 80% of this population chose to delay their treatment themselves [35]. This is another instance where fear of exposure to the virus influenced decisions made by patients for medical intervention. 

A further evolving area is the use of COVID-19 vaccinations in patients with cancer. Given the higher mortality rates from SARS-CoV-2 infection seen in these patients, the establishment of a safe and effective vaccine is crucial. Recent reports have suggested that COVID-19 vaccination for cancer patients is safe, immunogenic, and effective in protecting against symptomatic disease. However, the rates of seroconversion and protection offered by vaccination are generally lower in patients with cancer compared with the general population. Patients with hematological malignancies appear to be at particular risk, especially those who have received B-cell-depleting agents in the past 12 months [36]. Therefore, despite the relatively high coverage of COVID-19 vaccination programs in some countries, patients with cancer continue to remain at risk. Measures to address this include testing for antibody responses in cancer patients, and, in those found to have a deficient SARS-CoV-2 immune response, continuing to recommend adherence to infection-risk reduction measures and/or consideration of prophylactic treatment with COVID-19-neutralising monoclonal antibodies [37].

## 5. Impact on Clinical Trials

The effects of the pandemic on clinical trials were felt throughout administrative and clinical practices. Clinical trial initiation and recruitment were typically halted at the beginning of the pandemic as infection control measures were implemented [38]. Furthermore, trials of investigational medical products (IMPs) known to be immune-suppressive, or cause pneumonitis, were ceased due to concerns about the potential to exacerbate the severity of SARS-CoV-2 infection [39]. In a March 2021 report, a 60% decrease in new oncology clinical trials during the pandemic was noted, fueling concerns regarding slowing the development of new cancer therapies [40]. A major UK-based cancer research charity, Cancer Research UK, halted recruitment in 95% of their clinical trials, both to protect cancer patients, and to cope with the redeployment of clinical research staff to support frontline COVID-19-facing healthcare services. However, it is noteworthy that trials for pediatric and adolescent patients were affected to a lesser extent, following observations that young adults, children, and infants were less likely to develop severe SARS-CoV-2 infection and transmit it to others [41]. By halting start up and further trial recruitment, resources were prioritized to patients already enrolled in existing trials, who were benefiting from treatment and continued where possible. A further consideration was that the extensive monitoring and face-to-face physical assessments characteristic of normal clinical trial practice were no longer considered appropriate given the risks of exposure to the virus, either in hospital or during travel to and from the healthcare institution. This called for dramatic changes in routine practices and trial protocols to prevent the spread of COVID-19 while preserving trial integrity and patient welfare [42]. 

In the UK, the Medicines and Healthcare Products Regulatory Agency (MHRA) released guidelines in accordance with government restrictions to guide clinical trial management [43]. The shift in clinical trial protocol began with the scrutinization of every in-person appointment, with the aim of determining whether face-to-face assessment was necessary, or whether alternative strategies could be implemented to achieve the same objectives [44]. For example, standard physical examinations of marginal benefit such as weight, height, and blood pressure measurements, were typically excluded to reduce the need for hospital visits. Furthermore, screening tests such as echocardiograms and multi-gated acquisition (MUGA) scans were dispensed with in patients without risk factors or a history of cardio-vascular disease, as the risk of exposure to patients and healthcare workers could not be justified [45]. Other measures included the posting of informed consent (IC) forms and other documentation to patient’s homes instead of them being signed during in-person consultations; the option to e-sign documents was also incorporated and recognized as an accepted form of consent by regulatory organizations [44]. These bodies also typically encouraged the switch to IMPs in oral form where appropriate, as the IMP could be delivered directly to the patient’s home and be self-administered at the agreed schedule, reducing the need for them to travel to the trial site [43].

Telemedicine was also widely embraced as the best mode of communication to deliver the essential information to patients in remote video consultations, reducing the need for an in-person visit [44,46]. Notably, a systematic review has also provided evidence that supportive care and counselling delivered by telephone can be more convenient and noninferior to standard care for all outcomes, including knowledge, decision conflict, cancer distress, and perceived stress [47]. However, varying levels of computer literacy among trial participants must be taken into consideration. Pre-pandemic, one of the major obstacles faced by telemedicine was termed the “digital divide”, as there was a substantial disparity in IT literacy between elder and younger demographics [48]. Unsurprisingly, the most recent report highlighted this disparity, as data showed that 54% of adults aged 75 and over did not use the internet compared to 99% of individuals aged 16–44 years [49]. Despite this, the pandemic has resulted in a wider uptake of digital technology by older people, as meeting in person has not been feasible [46]. Study visits that could be conducted remotely were identified and adapted, such as the reporting of adverse events or suspected adverse events by video consultation or questionnaires/patient diary cards. Furthermore, some protocols also allowed for physical assessments and tests that would normally occur at the trial site to be carried out locally by primary care physicians and other health professionals [50]. Where these assessments could not be performed locally due to the nature of the assessment, the interval between assessments was increased to reduce the total number of patient visits to hospital. In addition, in-person tests such as imaging scans and follow-up sample collections were planned to occur within one visit to further minimize risk of exposure to the virus, which also inadvertently improved the patient experience [51]. Remote monitoring systems were also implemented, allowing patients to be followed-up from their homes. For example, the use of wearable devices that provide a stream of real time data (e.g., blood pressure/heart rate monitors) was also expanded. This is a welcome innovation, as these approaches have the advantage that the data can be collected over extended periods of time and in a more natural and relaxed environment, mitigating issues such as “whitecoat hypertension” [52,53].

A further demonstration of clinical trial modification during the COVID-19 pandemic, albeit outside of cancer research, was the Randomised Evaluation of Covid-19 Therapy (RECOVERY) trial. The RECOVERY trial took on an unorthodox approach to set up a large-scale randomized clinical trial using a digital portal. DigiTrials is a data platform that consolidates patient data across the UK health system, which made the patient population in critical care units across the UK more accessible for recruitment into the trial. The DigiTrial system also provided a universal platform for data collection and processing, which allowed for quicker analysis and reporting of results [54]. The RECOVERY trial took a flexible approach with its protocol, which allowed for trial arms to be added as new research unveiled potential COVID-19 treatment modalities, while also allowing for arms to be halted as soon as data (which was being analyzed on an ongoing basis) determined that the therapy was ineffective (e.g., the anti-malarial drug Hydroxychloroquine) [55]. This also made clinical trial management less arduous for healthcare workers who were already under extreme pressure. This was reflected in the speed of the results: the trial established that dexamethasone reduced the number of deaths by one-third in ventilated critical care patients in just 98 days [56]. 

## 6. Lessons Learnt for the Future of Oncology Clinical Trials

The RECOVERY trial and other trials have served as an example demonstrating how a flexible protocol, the use of a data platform, and integration with standard care can considerably improve trial efficiency and turnaround time. With a simplified consent form and a carefully thought-out “trial by design” approach, the RECOVERY trial astoundingly received regulatory approval within 9 days, which is drastic compared to the average 30–60-day time frame. Many health research authorities are still currently fast-tracking ethical approval for COVID-19 research and have managed to bring the normal 60-day cycle down to just 10 days. Another paradigm shifting example has been the development of SARS-CoV-2 vaccines. Prior to the pandemic, the timeline from basic science and target identification through the phases of clinical development and licensing typically took 10–20 years, a time not inconsistent with the time it takes to develop a new anti-cancer drug. However, effective vaccines for COVID-19 were developed, tested, and licensed within a year of the virus first being discovered [57,58,59]. Undoubtedly, there were several factors that contributed to this, with the global nature of the pandemic meaning that pharmaceutical companies, research institutions, and governments made/had access to massive resources available for this purpose. However, this period established the proof of principal that effective treatments can be safely developed in a fraction of the time that it has historically taken.

The acceleration of drug development aside, there are many other examples of innovative practice that have emerged during the pandemic, that can be further adapted and taken forward. As discussed above, the intensity and frequency of monitoring assessments for patients on study was streamlined during the pandemic, with the aim of avoiding exposure to the virus [44]. As it now seems that many of these visits are unnecessary, future protocols should be designed with reduced numbers of visits to improve efficiency and patient experience, if retrospective analyses show no impact upon trial integrity [60]. This includes repeat biopsies and venepunctures, as it has been observed that the reduction in sample collection during the pandemic inadvertently contributed to an improved patient experience. Reducing the intensity of monitoring also reduces the inconvenience of traveling, which in turn helps address existing issues with patient retention in clinical trials. Patient drop-out and insufficient recruitment are often the cause of trials failing to complete accrual within the study period [61]. This is frustrating from a clinical perspective, as it prolongs the approval of beneficial therapies to reach a wider patient population. However, from a pharmaceutical and clinical research organization (CRO) perspective, failing to complete trials within aliquoted time frames can have financial implications and tarnish the reputation of the investigators. The cost to complete clinical trial research is then reflected in the costing of the therapy, driving up prices of new therapies. Ethical and regulatory standards in clinical trials have effectively managed risk and protected against litigation, at the expense of increased bureaucracy. Paradoxically, these stringent measures enforced to maximize patient safety and prevent malpractice have a wider negative impact on patients in standard care, as the elongation of trials/trial initiation ultimately delays the accessibility of more efficacious therapies. 

Many healthcare systems are permanently under a degree of stress, which was experienced even before the COVID-19 pandemic, both financially and due to a shortage of appropriately trained staff. Efforts should be made to integrate trial design more closely with standard hospital workflow to lessen the burden on health services, promote an increase in clinical trial activity, and to develop resilience so that clinical trials can continue, should there be future waves of COVID-19 or another novel infectious agent [50]. One advantage of this is that if the clinical trial experience mimics the normal standard of care, then this should improve the quality of clinical trial data. A common problem is that the results of major clinical trials are not always recapitulated in the “real-world” setting, in part due to patient selection, but also due the increased monitoring and physical and psychological support given to the participants. If clinical trials resemble standard care more closely, then disparities between clinical trial and “real-world” data are less likely to be as apparent. The pandemic also offered significant training opportunities as staff were redeployed. For example, in some centers, oncology research nurses were redeployed to assist with COVID-19 vaccination trials, due to their experienced research nurses in trialing IMPs with significant side effect profiles [62]. Standardized “cross-training” of clinical research staff would allow for interchangeability of research staff between different studies. This would reduce staffing burden and potential burn-out in the event of unforeseen circumstances that result in clinical staffing shortages when running trials [60].

There have been many examples for how the administrative and regulatory burden imposed on research sites can be improved. New policies were implemented by CROs, sponsors, and regulatory authorities during the pandemic that facilitated a magnitude of administrative duties surrounding recruitment, site selection, and study monitoring by leveraging technology and simplifying reporting processes. This included implementing virtual online meeting platforms (i.e., Zoom/Microsoft Teams) and protected e-DocuSign (via AdobeSign) to support remote work for many administrative employees for a reduced dependence on physical paperwork. Standard practice in clinical trials requires all protocol violations to be thoroughly documented within the details of the trial to allow for accurate analysis of the data. The deviation reporting process during the pandemic was scaled back to combat the surge of administrative work because of the several protocol violations put in place to adapt trials. Only major deviations were reported in stand-alone deviation reports and all minor protocol deviations were reported on a weekly basis collectively, opposed to raising a report per deviation, whether it be major or minor [63]. Furthermore, there is considerable variability in the methods of how protocol deviations and trial amendments are documented across different CROs, which makes comparative analysis between studies much more complex. Standardizing how deviations are recorded, potentially using an inter-operator technology platform, would reduce administrative paperwork, allowing protocol violations to be reported in a more accessible and consistent manner. Accessibility of patient records and medical notes remotely was a significant issue faced by clinical trial monitors during the pandemic. Patient information is strictly confidential, which limits the modes it can be shared as this raises several issues with general data protection regulation (GDPR). The digitalization of data collection during the pandemic played a huge role in delivering rapid trial results. A similar approach should be adopted for the recording and processing of trial data on a secure and regulatory-approved digital platform, which anonymizes patient information and facilitates the remote access of trial data. As modern trials have become progressively complex with increases in the sophistication of therapy, it seems reasonable that the nature of regulation and administration should evolve as well.

## 7. Conclusions

The impacts of the COVID-19 pandemic on oncology care and clinical trials have been immense, and there is no doubt that the pandemic will leave a permanent impression on healthcare services. As devastating as the disruption has been, the response to the pandemic has seen many innovations and adaptations that will continue to be of great value in the future [64] (Figure 2). The pandemic has provided an opportunity to address some of the inefficiencies in the conduct of clinical trials, such as excessive bureaucracy and patient monitoring, and has driven improvements that offer the potential for future trials to be carried out more quickly and safely. Major successes include the rapid set up of COVID free cancer hubs that facilitated the provision of anti-cancer drug, radiotherapy, and surgery could continue where possible, with a reduced risk of virus transmission In addition, the swift implementation of telehealth platforms as a form of virtual meetings between patients and health professionals, as well as meetings between colleagues and professionals from other organizations within healthcare, has been a substantial leap in the maturation of digitalizing healthcare services [65]. However, the notion of solely relying on telehealth platforms post pandemic is far from reality. It is most likely that telehealth will become an integrated part of patient−doctor communication as a means to improve working efficiency and patient convenience. However, it is probable that the option for face-to-face consultations will still be available in consideration of the issue of digital exclusion, particularly among the elderly population where low computer literacy is most prominent [66].

In clinical trials, the COVID-19 pandemic caused a momentary pause that necessitated a step back to begin initiating change on very complicated and established practices. In a matter of months, the clinical trial industry has made a huge leap in leveraging technology that may have taken years to implement into normal practices without these drivers. The successful digitalization of many administrative processes has opened the door for the application of innovative technologies and methodologies to improve working efficiency. In the longer term, this offers the potential to reduce the burden of clinical documentation and increase the speed of data acquisition and interpretation, which should also result in a faster translation to the clinic. The experience from the pandemic should also inform the development of new measures to make healthcare systems and clinical trial networks more resilient to future healthcare crises, such as further waves of COVID-19, novel infectious agents, or other challenges. 

## Figures and Tables

**Figure 1 cancers-13-05924-f001:**
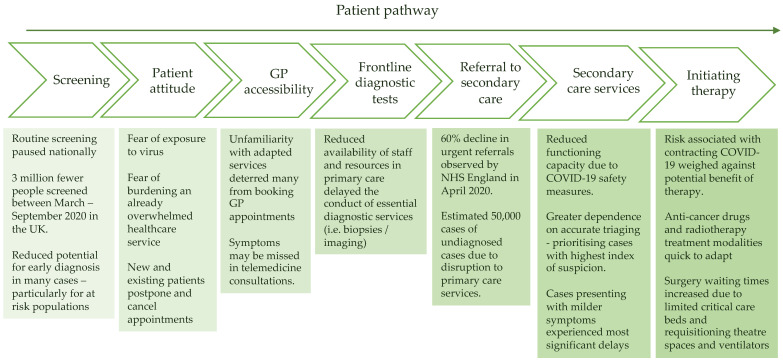
Impact of COVID-19 on the oncology care pathway.

**Figure 2 cancers-13-05924-f002:**
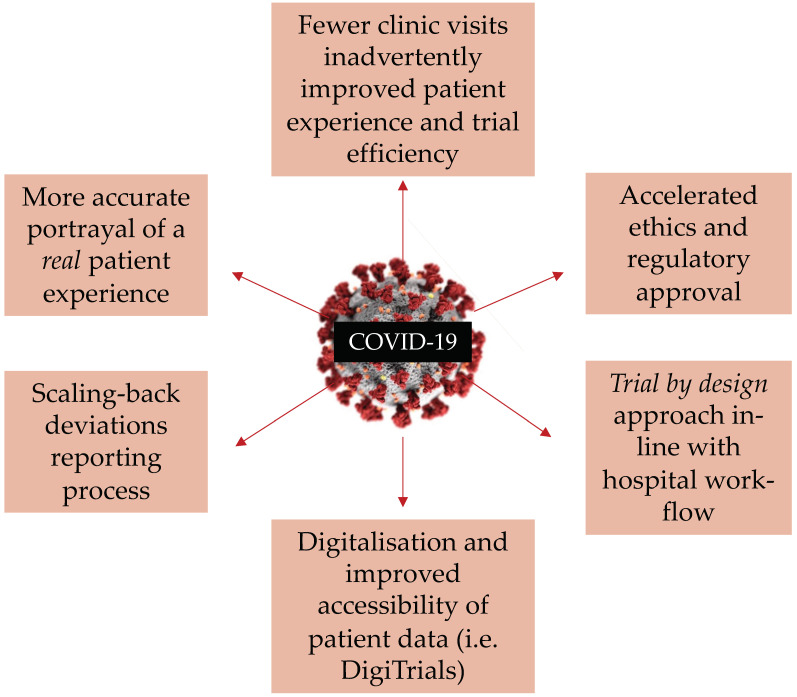
Potential positive impacts on oncology clinical trials due to COVID-19.

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
