# Peer review of "The Impact of the COVID-19 Pandemic on Oncology Care and Clinical Trials"

_cancers, 2021, doi:10.3390/cancers13235924_

Round 1

Reviewer 1 Report

This is a timely, interesting narrative review regarding the multidimensional impact that the COVID-19 pandemic posed on oncology patients and healthcare workers. The authors describe in detail, how the measures/restrictions regarding the containment of the SARS-CoV-2 pandemic affected both initial diagnosis as well as initial treatment of patients diagnosed with malignancy. Moreover, they describe the strategies/approaches that many oncology centers implemented in order to both protect patients from COVID-19 while also providing the appropriate medical care. 

Simple summary & abstract: Please avoid repetition of it will discuss

Introduction: Authors may include more citations/references in the first paragraph lines 33-45.

Figure 1 should be revised with a more readable figure for the reader

2 --> Lines 77-110: These two paragraphs describe similar issues and could be potentially merged in one paragraph. 

4 --> Authors describe in a detailed manner the different approaches and measures that were taken in order to minimize the potential exposure to SARS-CoV-2  whilst receiving the appropriate therapy. However, this section could be shortened.

5 --> the First paragraph could be omitted since it does not add a lot to an already detailed section.

Lines 257-283 Authors should also describe the positive effect that due to COVID restrictions, multidisciplinary teams manage to combine many interventions (e.g. radiotherapy, chemotherapy, imaging studies) in one appointment which is really important from a patient's perspective.

The last paragraph of section 5 and 1st paragraph of section 6 should be combined in one paragraph.

Figure 2 should also be updated in a more readable version.

Author Response

The impact of the COVID-19 pandemic on oncology care and clinical trials

Jennyfa K. Ali; John C. Riches

Comments from Reviewer 1 with our responses in italics

This is a timely, interesting narrative review regarding the multidimensional impact that the COVID-19 pandemic posed on oncology patients and healthcare workers. The authors describe in detail, how the measures/restrictions regarding the containment of the SARS-CoV-2 pandemic affected both initial diagnosis as well as initial treatment of patients diagnosed with malignancy. Moreover, they describe the strategies/approaches that many oncology centers implemented in order to both protect patients from COVID-19 while also providing the appropriate medical care. 

We thank the Reviewer for their kind comments and excellent suggestions regarding the manuscript. The following changes have been made to the manuscript to reflect their suggestions:

Simple summary & abstract: Please avoid repetition of it will discuss

We have re-written the simple summary and abstract to avoid repetition of this phrase.

Introduction: Authors may include more citations/references in the first paragraph lines 33-45.

We have included two more references in this section.

Figure 1 should be revised with a more readable figure for the reader

We have revised figure 1 to enhance readability by reducing the amount of text and increasing the font size.

2 --> Lines 77-110: These two paragraphs describe similar issues and could be potentially merged in one paragraph. 

We have edited these two paragraphs to remove any duplication of content.

4 --> Authors describe in a detailed manner the different approaches and measures that were taken in order to minimize the potential exposure to SARS-CoV-2 whilst receiving the appropriate therapy. However, this section could be shortened.

We have edited this section to shorten it.

5 --> the First paragraph could be omitted since it does not add a lot to an already detailed section.

We have removed this paragraph as suggested.

Lines 257-283 Authors should also describe the positive effect that due to COVID restrictions, multidisciplinary teams manage to combine many interventions (e.g. radiotherapy, chemotherapy, imaging studies) in one appointment which is really important from a patient's perspective.

We have added a phrase to this paragraph to emphasise the important impact on patient experience.

The last paragraph of section 5 and 1st paragraph of section 6 should be combined in one paragraph.

We have rewritten these two paragraphs to reflect this suggestion.

Figure 2 should also be updated in a more readable version.

We have also reformatted this figure to improve clarity for the reader.

Reviewer 2 Report

Dear authors,

The manuscript entitled “The impact of the COVID-19 pandemic on oncology care and clinical trials” has an important role for cancer patients and this review will add information to the readers in the oncology treatment care in this new era of COVID-19 pandemic. The flexibilities used during pandemic are very well described to minimize the effects for this special population.

The manuscript is very well written, but there are few comments below that could improve the manuscript and the information to the readers.

Line 44-45: Authors describe the concern about the high risk of cancer patients were for severe disease/death if they have COVID-19, but they did not mentioned the mortality rates. There are few studies about it (DOI: 10.1007/s10147-021-01863-6; doi: 10.1016/S0140-6736(20)31173-9); it will be add more information for the readers if they know how people with cancer were affected by COVID-19 pandemic.

The authors also are describing how the fast approval of clinical trials could improve the cancer healthcare. What are they know about the trials for vaccines in these special population? It is also important to reveal if the cancer patients are willing to receive vaccines to return their activities for therapy purposes. Other important point is related to the efficiency of the vaccination for this special population, especially for those who are under cancer treatment. Is the mortality rate going down in this population after vaccination? It is another impact that should be included in this manuscript, since vaccine is now a reality for several countries in the world.

Author Response

The impact of the COVID-19 pandemic on oncology care and clinical trials

Jennyfa K. Ali; John C. Riches

Comments from Reviewer 2 with our responses in italics

The manuscript entitled “The impact of the COVID-19 pandemic on oncology care and clinical trials” has an important role for cancer patients and this review will add information to the readers in the oncology treatment care in this new era of COVID-19 pandemic. The flexibilities used during pandemic are very well described to minimize the effects for this special population.

The manuscript is very well written, but there are few comments below that could improve the manuscript and the information to the readers.

We thank the Reviewer for their kind comments and excellent suggestions regarding the manuscript. The following changes have been made to the manuscript to reflect their suggestions:

Line 44-45: Authors describe the concern about the high risk of cancer patients were for severe disease/death if they have COVID-19, but they did not mentioned the mortality rates. There are few studies about it (DOI: 10.1007/s10147-021-01863-6; doi: 10.1016/S0140-6736(20)31173-9); it will be add more information for the readers if they know how people with cancer were affected by COVID-19 pandemic.

We have added the mortality rate data from these publications to the manuscript.

The authors also are describing how the fast approval of clinical trials could improve the cancer healthcare. What are they know about the trials for vaccines in these special population? It is also important to reveal if the cancer patients are willing to receive vaccines to return their activities for therapy purposes. Other important point is related to the efficiency of the vaccination for this special population, especially for those who are under cancer treatment. Is the mortality rate going down in this population after vaccination? It is another impact that should be included in this manuscript, since vaccine is now a reality for several countries in the world.

We agree that the role of COVID-19 vaccinations in patients with cancer is particularly important. We have added a paragraph discussing this in section 4.

Reviewer 3 Report

The manuscript titled "The impact of the COVID-19 pandemic on oncology care and clinical trials" submitted to Cancers, describes a very interesting and very important topic related to the impact COVID-19 on oncology clinical trials. The topic is actual not only in EU but also all over the worldReview is very well organized and written and it reads very well. Manuscript is divided properly and after a well-presented and comprehensive introduction, authors described impact on the initial presentation of patients with potential cancer, on the initial presentation of patients with potential cancer, on the initial presentation of patients with potential cancers and finally on clinical trials.  Authors included also comprehensive paragraph catchy titled: Lessons learnt for the future of oncology clinical trials?.  To sum up, review is written very comprehensive way. The authors properly concluded at the conclusion that  the COVID-19 pandemic caused a momentary pause which necessitated a step back to begin initiating change on very complicated and established practices. Additionally according authors statement the experience from the pandemic should also inform the development of new measures to make healthcare systems and clinical trial networks more resilient to future healthcare crises, such as further waves of COVID-19, novel infectious agents or other challenges.

Author Response

Our response to Reviewer 3:

We thank the reviewer for their kind comments regarding the manuscript